# Novel Strategies for the Bioavailability Augmentation and Efficacy Improvement of Natural Products in Oral Cancer

**DOI:** 10.3390/cancers15010268

**Published:** 2022-12-30

**Authors:** Alisha Sachdeva, Dimple Dhawan, Gaurav K. Jain, Mükerrem Betül Yerer, Taylor E. Collignon, Devesh Tewari, Anupam Bishayee

**Affiliations:** 1Department of Pharmaceutics, Delhi Institute of Pharmaceutical Sciences and Research, Delhi Pharmaceutical Sciences and Research University, New Delhi 110 017, India; 2Center for Advanced Formulation Development, Delhi Pharmaceutical Sciences and Research University, New Delhi 110 017, India; 3Department of Pharmacology, Faculty of Pharmacy, Erciyes University, Kayseri 38039, Turkey; 4College of Osteopathic Medicine, Lake Erie College of Osteopathic Medicine, Bradenton, FL 34211, USA; 5Department of Pharmacognosy and Phytochemistry, School of Pharmaceutical Sciences, Delhi Pharmaceutical Sciences and Research University, New Delhi 110 017, India

**Keywords:** cancer, oncogenes, oral cavity, natural products, nanoparticles

## Abstract

**Simple Summary:**

Oral cancer is a major cause of death worldwide. The major challenges associated with the conventionally used treatment modalities include the recurrence of the cancer and the adverse effects of the chemotherapeutic drugs. Natural products play a very important role in drug discovery and have potential effects against various cancers in general and oral cancer in particular. The major problem with natural products is their poor bioavailability, which limits their therapeutic application. Therefore, an attempt is made to comprehensively utilize various formulation strategies that can help in improving the bioavailability of various anticancer natural compounds. In this work, we have presented recent advancements in novel strategies for natural product delivery that lead to the significant enhancement of bioavailability, in vivo activity, and fewer adverse events for the prevention and treatment of oral cancer.

**Abstract:**

Oral cancer is emerging as a major cause of mortality globally. Oral cancer occupies a significant proportion of the head and neck, including the cheeks, tongue, and oral cavity. Conventional methods in the treatment of cancer involve surgery, radiotherapy, and immunotherapy, and these have not proven to completely eradicate cancerous cells, may lead to the reoccurrence of oral cancer, and possess numerous adverse side effects. Advancements in novel drug delivery approaches have gained popularity in cancer management with an increase in the number of cases associated with oral cancer. Natural products are potent sources for drug discovery, especially for anticancer drugs. Natural product delivery has major challenges due to its low solubility, poor absorption, inappropriate size, instability, poor permeation, and first-pass metabolism. Therefore, it is of prime importance to investigate novel treatment approaches for the delivery of bioactive natural products. Nanotechnology is an advanced method of delivering cancer therapy with minimal damage to normal cells while targeting cancer cells. Therefore, the present review elaborates on the advancements in novel strategies for natural product delivery that lead to the significant enhancement of bioavailability, in vivo activity, and fewer adverse events for the prevention and treatment of oral cancer. Various approaches to accomplish the desired results involve size reduction, surface property modification, and polymer attachment, which collectively result in the higher stability of the formulation.

## 1. Introduction

Oral cancer (OC) is a highly prevalent disease in India, representing 50–70% of all cancer-related deaths, and has the greatest incidence in Asian nations [1]. The anterior tongue, cheek, gingiva, mouth floor, and other regions of the oral cavity can be affected in OC. The prevalence of oral cavity cancer varies greatly across the globe. Only 5% of all malignancies in the United States, Western Europe, and Australia are due to OC. Some of the countries with the greatest rates of OC worldwide include India, a few regions of Brazil, France, and Central and Eastern Europe. There has been an association between the use of alcohol and tobacco products and increased rates of OC in France and Eastern Europe [2]. The primary cause of OC in India is believed to be chewing betel nut leaves that have been wrapped with lime and tobacco or pan, bringing the buccal mucosa in constant contact with the cancer-causing agent, such as tobacco. The age of an individual is proportionally related to OC prevalence. Between the ages of 40–49, incidence rates spike substantially, peaking around ages 70 to 79. OC is more prevalent among men, and, depending on where it develops in the oral mucosa, men have a two- to six-fold increased risk of developing the condition. This is mainly because men consume more alcohol and cigarettes than women do [2].

Cancers of the lip, oral cavity, hypopharynx, oropharynx, and larynx are known collectively as mouth and oral cancers. Of these, cancers of the lip and oral cavity are the most common, with more than 377,700 cases worldwide in 2020 [3]. OC is a head and neck malignancy affecting the tongue, lips, alveolar mucosa, mouth floor, buccal mucosa, gingiva, palate, or a combination of these areas [4]. Oral squamous cell carcinomas (OSCC) account for 90% of OC cases [5]. Alcohol use (particularly when associated with smoking) and smoking have both been identified as additional risk factors for most malignancies [6]. Cigarette smokers are five to twenty-five times more likely to develop cancer than nonsmokers, with data pointing to a dose-dependent relationship. Around 45–72% of the patients with OC worldwide survive the disease for five years [7]. Multiple variables, acting alone or in combination synergistically, cause the buccal epithelium to become permanently damaged during oral carcinogenesis. OC results from a number of genetic modifications that develop gradually with time, as well as the ability of the tumor to bypass the host immune function [8,9]. Precancerous changes such as leukoplakia, erythroplakia, or oral submucous fibrosis prelude tumor growth and result from the epithelium undergoing a malignant transition. These findings [10] may be malignant but are typically asymptomatic and overlooked; therefore, the majority of cases are diagnosed in their later stages [11].

OSCC is amongst the most challenging conditions to resolve due to its propensity to metastasize to regional lymph nodes, invade local tissue, and develop resistance to chemotherapy agents. These factors contribute to unpredictable prognoses and unfavorable outcomes. Around 90% of post-radiotherapy and surgery treatment failures are due to local recurrences [12,13]. The general prognosis of OSCC has not improved despite significant therapeutic advancements, involving surgical techniques and adjuvant treatments, indicating the urgent need for a novel OSCC treatment approach [14].

The term “oral cavity” (Figure 1) refers to the anatomical region bound by an imaginary coronal plane from the hard and soft palate junction superiorly, to the circumvallate papillae of the tongue inferiorly, to the vermillion of the lips anteriorly [15]. Mastication, deglutition, oral competency maintenance, and speech articulation are the four primary functions of the cavity. In order to ensure both a high quality of life and a decent chance of human survival, several criteria must be taken into account when selecting the cancer treatment modality, and the therapy provided should always be adapted specifically to each patient’s wishes [16,17]. Partial glossectomy can be used to treat small tumors (stage T1 and early-T2 tumors, i.e., tumors 3 cm in diameter), provided the procedure does not significantly affect the mouth floor. The defect can be mostly closed with an operationally efficient and normal-shaped tongue after a partial glossectomy when done in a horizontal spindle approach. Hemiglossectomy, partial glossectomy, or total glossectomy may be needed for large tumors. A radial forearm free flap can be used to rebuild hemiglossectomy deficits and restore speech and swallowing abilities [17,18].

Natural products are potent sources for the discovery of drugs, including those used for cancer therapy [19,20,21]. Natural products are an emerging approach for the prevention and treatment of head and neck cancers, including tobacco-related OC [22,23,24]. Bioactive natural products were reported to be efficacious in OSCC [25,26]. The United States National Cancer Institute claims that the majority of antitumor medications are made from natural sources [27]. Some of the most common anticancer treatments, such as vinblastine, vincristine, and paclitaxel, are derived from plants [28,29,30]. An interesting approach to cancer therapeutic research is the study of plants, fungi, and algae as carriers of novel chemical substances with chemopreventive properties. Natural remedies are more affordable, less toxic, and less harmful when compared to standard chemotherapeutic approaches [31,32,33]. The phytonutrients isolated from natural materials may selectively affect tumor cells without damaging healthy cells. Phytochemicals are regarded as having outstanding possibilities for the development of anticancer medicines because of their pleiotropic influence on target events happening during oral carcinogenesis and participation in a number of signaling cascades. Because of their low toxic effects, safety, and accessibility, natural products are of paramount importance and present an accessible solution to the current difficulties with modern cancer therapies [34,35]. Natural substances, including phytochemicals, have shown anticarcinogenic activity in the development and spread of cancer by altering the characteristic framework of carcinogenesis recognized as “the hallmarks of cancer” [11,36,37].

The use of phytochemicals for chemoprevention is gaining popularity as it is thought to be a low-cost, easily implementable, acceptable, and available method for the treatment and management of cancer. For the purpose of cancer chemoprevention, several phytochemicals are undergoing preclinical or clinical research. According to epidemiological research, diets abundant in fruits and vegetables reduce the risk of developing cancer [34].

Some of the biologically active components in natural products have poor distribution and absorption, which reduces their bioavailability and effectiveness and may make them less useful for clinical use. Several nanomedical strategies (Figure 2), including liposomes, microemulsions, solid lipid nanoparticles (SLNs), polymeric NPs, liquid crystal systems (LC), and precursor systems for liquid crystals (PSLCs), have been suggested to avoid these obstacles in the formulation of herbal medicines, food supplements, and essential oils. Biotechnological approaches may help to improve the bioavailability and bioactivity of herbal medication preparations because NPs have been utilized to change and improve several drugs’ pharmacokinetic and pharmacodynamic properties [38,39].

Natural products have been reported as potential candidates for anticancer drug delivery, with improved pharmacokinetic properties and an enhanced cellular uptake [39,40,41,42]. Therefore, it is of prime importance to investigate alternate OC treatment approaches using novel formulations and delivery systems for bioactive natural agents. Nanotechnology is an advanced method of delivering cancer therapy with minimal damage to normal cells while targeting cancer cells. To the best of our knowledge, no reviews have comprehensively discussed the novel drug delivery strategies for the natural products used in OC. Therefore, the present review elaborates on the advancements in novel strategies for natural product delivery that lead to the significant enhancement of bioavailability, in vivo activity, and fewer adverse events for the prevention and treatment of OC.

## 2. Epidemiology and Etiology of Oral Cancer

Sri Lanka, India, Pakistan, Bangladesh, Hungary, and France have the highest rates of oral cancer [43]. An estimated 66,650 additional cases are reported in the European Union each year. The two main causative factors for small cell carcinoma (SCC) of an oral cavity are alcohol and tobacco use [44]. The Asian population has also been linked to other harmful behaviors, including smoking and consuming betel nuts. Numerous cancer-causing chemicals, particularly polycyclic hydrocarbons and nitrosamines, are found in tobacco. Oropharyngeal and oral SCC appear to be caused by a synergistic interaction between alcohol and tobacco [44,45,46,47]. Other factors have been identified as causative factors, including poor dental hygiene, exposure to wood dust, nutritional deficits, and ingestion of salted and red meat [48,49].

An indication as to the origin of the particular tumor may be provided by the sequence of the specific genetic mutation in OC patients. In head and neck squamous cell carcinoma (HNSCC), [50] those who had smoked and consumed alcohol had a significantly higher incidence of the p53 mutation than non-users [2].

For 75% of all instances of squamous cell carcinoma of the head and neck (SCCHN), alcohol and cigarette use are the most often recognized etiologic variables [51]. More nationally representative evidence from the Million Death Study revealed that the age-standardized death rate was 22.1 for head and neck cancer (HNC) [52]. The significance of OCC in public health is that the majority of the afflicted individuals come from lower socioeconomic groups, which are more likely to be exposed to risk factors, including cigarette smoking, especially in India, Bangladesh, and Pakistan. Additionally, access to qualified healthcare professionals and services is restricted for patients living in rural parts of middle- and low-income nations [8]. Many cases present at advanced stages of the disease due to delays in detection and referrals, resulting in lower survival rates and increased patient expenses [53,54,55].

There have been several risk factors or potential causes of OC reported. It has been demonstrated that OC is greatly influenced by chemical factors, such as alcohol and tobacco use, biological factors, including the human papillomavirus (HPV), syphilis, oro-dental factors, nutritional deficits, and viruses [2].

There are numerous indications that tobacco use, in all forms, including smoking, chewing, betel quid, and others, has a cancer-causing effect on the oral cavity. Smoking is the most prevalent tobacco usage method [2]. Alcohol has also been described in a number of articles as a significant risk factor for OC. According to studies, people who consume over 170 g of whiskey per day are 10 times more likely to develop OC than light drinkers [56]. Additionally, it has been hypothesized that alcohol makes it easier for carcinogens to enter exposed cells, changing the way oral mucosal cells function [57]. However, the available research does not suggest that pure ethanol is a carcinogen stimulating the growth of OC directly [2].

Apart from the chemical factors, various biological factors are also attributed to OC risks. The cellular machinery of the host can be taken over by viruses, which can also mutate DNA and chromosomal structures and cause changes in cell proliferation. Herpes simplex virus (HSV) and human papillomavirus (HPV) [58] have both been identified in recent years as OC-causing pathogens. Around 23.5% of OC patients have had HPV [59] detected. It is unclear how HPV in pre-cancerous oral lesions affects the prognosis. A few studies, however, have discovered greater disease-specific survival and a better prognosis for OC that is positive for HPV. HSV-1, also known as “oral herpes”, is frequently linked to lesions on the lips and mouth and may be a possible cause of OC [60].

*Candida* (a type of fungus responsible for candidiasis) may also contribute to the onset of OC. According to clinical investigations, nodular leukoplakia that is infected with Candida tends to have a greater rate of dysplasia and a higher propensity to become malignant. Additionally, it has been demonstrated that chick embryo epithelium exhibits squamous metaplasia and a greater proliferative phenotype after *Candida albicans* infection [61]. The relationship between *Candida* infection and OC needs further investigation before any associations can be concluded.

Oral hygiene and a lack of certain nutrients and minerals, such as carotenoids, antioxidants, phenolic compounds, terpenes, steroids, indoles, and fibers, increase the chance of developing cancer. This is due to the inadequate dietary consumption of vegetables and fruits. These foods have phytochemicals, which are bioactive substances with protective properties. Oral disorders are thought to be facilitated by a deficiency in phytochemicals [62,63]. Betel nut and areca nut have been identified by the International Agency for Research on Cancer (IARC) as category I human carcinogens in addition to being psychostimulants and addictive drugs. Areca nuts’ primary active ingredients include betel quid alkaloids and polyphenols, both of which have been linked to oral and pharyngeal cancer [64]. Alkaloids in betel quid are thought to be the areca nut’s “active component”, and arecoline and arecaidine are the two most poisonous components. Arecoline and arecaidine are mutagens that can cause sister chromatid exchange, chromosomal abnormalities, DNA strand breakage, and the development of micronuclei in mammalian cells [65].

## 3. Pathophysiology of Oral Cancer

### 3.1. Pathogenesis

The squamous mucosa of numerous head and neck structures can develop leukoplakia, erythroplakia, and leukoerythroplakia, all of which are noninvasive lesions. These premalignant lesions develop a number of genetic mutations, which set off a chain of events that includes hyperplasia, dysplasia, and in situ and invasive cancer. Different phases of cancer development are linked to heterozygosity loss at certain chromosomes [66]. In this development model, the loss of 9p, which results in the inactivation of the p16 tumor suppressor gene, happens very early during the change from normal to hyperplastic mucosa [67]. The loss of 3p and 17p typically occurs after this genetic change in the transition to dysplasia [68]. The p53 gene is found on the 17p chromosome. Certain additional losses, such as 11q, 13q, 14q, 6p, 8, and 4q, result in the shift to invasive SCC, which is linked to the change from dysplasia to carcinoma in situ. In HNC, several cell cycle regulators such as cyclin D1, p16 CDK inhibitor, p53, and c-myc are also overexpressed. The carcinogenesis of HNC, including malignancies of the oral cavity, also involves cell surface receptor signaling pathways [69]. Additionally, there is now more interest in investigating cyclooxygenase-2 (COX-2) inhibitors either exclusively or in conjunction with inhibitors of the epidermal growth factor receptor (EGFR) in the context of chemoprevention due to the expression of COX-2 at various degrees of dysplasia [70]. EGFR as a key target for therapeutic medicines in aerodigestive malignancies may lead to advancements in cancer cell proliferation management [71].

Overexpression of EGFR, a receptor tyrosine kinase belonging to the ErbB family, has been seen in more than 80% of HNC [72]. Elevated EGFR expression was linked to poor survival in a meta-analysis that included information from over 6700 individuals with HNC [55,73]. Over 90% of all cases of OC are SCC. SCC formation has been linked to several premalignant lesions [74]. The probability for malignant development varies among the most prevalent premalignant lesions, including leukoplakia, erythroplakia, oral lichen planus, and oral submucous fibrosis [75]. The term “leukoplakia” refers to a white area, either a patch or plaque, that can’t be diagnosed as another disease [76]. Usually, drinking and smoking are correlated with this lesion. Less than 5% of oral cavity malignancies are minor salivary gland carcinomas, which usually develop on the hard palate (60%), lips (25%), and buccal mucosa (15%). The most typical classification (54%) is mucoepidermoid carcinoma, which is preceded by low-grade adenocarcinoma (17%) and adenoid cystic carcinoma (15%) [77,78,79].

### 3.2. Molecular Pathogenesis of Oral Cancer

The progression of oral carcinogenesis is similar to that of other cancers. The normal epithelium goes through many stages, beginning with dysplasia and terminating with invasive phenotypes. SCC is the most prevalent variety of OC, despite the fact that many kinds of carcinomas can be found in the oral cavity. In recent years, the molecular pathological description of OC has been unveiled through the use of genomic and proteomic approaches. There is ongoing research to determine the roles of genomic instability, epigenetic changes, and the generation of a gene expression pattern in the development of OC [80].

#### 3.2.1. Genetic Susceptibility

About 10% of all malignancies are known to have a significant genetic component. Studies demonstrating familial clustering [81] raise the possibility that there is a genetic component to the development of OC. Certain ethnic groups, such as the Ashkenazi group in Israel, have a clustering of OC, with a two-fold incidence relative to the rest of the Jewish population in the nation. Yet, it is still unclear what causes this hereditary predisposition. Current research has focused on examining particular genetic variations in vital genes implicated in oral carcinogenesis. For HNC, including OC, the glutathione S-transferase M1 (GSTM1) null genotype was reported as the most reliable polymorphism susceptibility marker. Tripathy and Roy’s meta-analyses revealed that the GSTM1 null genotype significantly elevated HNSCC risk by 20–50% [82].

#### 3.2.2. Proto-Oncogenes, Oncogenes, and Genetic Alterations

Point mutations, rearrangements, amplifications, and deletions are examples of genetic abnormalities that define the molecular basis of carcinogenesis. Oral carcinogenesis has also been linked to a number of oncogenes [80]. EGFR, c-myc, K-ras, int-2, parathyroid adenomatosis 1 (PRAD-1), and B-cell lymphoma (bcl)-like oncogenes have all been expressed abnormally during the development of OC [83]. In a 7,12-dimethylbenz[a]anthracene (DMBA)-induced hamster cheek pouch malignant OC model, overexpression and amplification of the cellular oncogene EGFR have been observed [84]. Neovascularization and mitogenesis are known to be facilitated by transforming growth factor-α (TGF-α) which has been demonstrated in human OC and hamster oral tumors [84].

#### 3.2.3. Tumor Suppressor Genes

The most frequent genetic alteration in all human malignancies is p53 inactivation [85]. A p53 mutation is present in over 50% of all primary HNSCC [50]. Chromosome 9p21–22 contains the HNC region that is most frequently deleted [69]. In most aggressive HNC tumors, chromosome 9p21 is lost [67]. The most frequent genetic mutation observed in this area is homozygous deletion. This deleted area contains p16 (CDKN2), a powerful cyclin D1 inhibitor [86]. The majority of advanced pre-malignant lesions have also been found to have lost the p16 protein on chromosome 9p21 [87]. When p16 or p16ARF are introduced into HNC cell lines, proliferation is effectively suppressed [88]. Most human cancers, including OC, frequently lose chromosome 17p. About 60% of malignant lesions exhibit it. Even though p53 inactivation and the loss of 17p are strongly correlated in invasive lesions, p53 alterations seem extremely uncommon in early lesions with 17p loss. Primary cancers have also been found to lose chromosomal arms 10 and 13q [89]. The fragile histidine triad (FHIT) tumor suppressor gene and a shared unstable site, FRA3B, are both located on chromosome 3p14. The deletion of FRA3B has been observed to occur often in early carcinogenesis [90] and is linked to cigarette smoking [91].

#### 3.2.4. Genomic Instability and Epigenetic Alterations

Genomic instability, such as the loss of heterozygosity (LOH) and microsatellite instability (MSI), is routinely detected in cancer. Methylation is the main epigenetic alteration in tumors. The development of tumors can be significantly influenced by modifications in methylation patterns. Epigenetic changes play a significant role in several essential genetic processes during carcinogenesis and are typically linked to the loss of genomic expression. Because these changes have the potential to deactivate DNA repair genes, malignant progression occurs. A methylation-specific polymerase chain reaction revealed unusual hypermethylation patterns in the p16, methylguanine-DNA methyltransferase (MGMT), and death-associated protein kinase (DAP-K) in smears of patients with HNC [92].

## 4. Conventional Treatments for Oral Cancer

Conventional therapies (CT) for OC target the cancerous mass through the use of chemotherapy, radiation therapy, and/or surgery, depending on the position and stage of the tumor. Despite the advantages, many therapies come with side effects, including functional loss, alterations to appearance, xerostomia, mucositis, and osteonecrosis of the jaw, as well as dental, hearing, thyroid, and ocular problems, among others [93]. Patients’ perceptions of CTs are often negatively impacted by the side effects of chemotherapy, including nausea, vomiting, loss of hair, and exhaustion, which reduce treatment compliance [94]. Therefore, OC therapy alternatives that lessen CT’s negative effects are needed [11].

Standard therapies based on stage, chemoradiation therapy, and induction chemotherapy are the conventional treatments for OC [95]. For the treatment of OC in modern therapy, anticancer medications such as cisplatin, fluorouracil, cetuximab, paclitaxel, methotrexate, and docetaxel (DTX) have been used either solely or in combination [96,97,98,99]. Traditional cancer therapy approaches for solid and malignant tumors, such as surgery, are frequently used as the initial course of therapy. Various surgical procedures, including radical or curative surgery (complete tumor resection), surgery for symptomatic treatment, conserving surgery, surgery for metastases, recurring surgery, and reconstructive surgery are used to treat various stages of cancer. Different surgical techniques have been created as a result of technological breakthroughs, which might eradicate the necessity for invasive surgeries given their tendency to be pricey, be painful, and carry a higher risk of infection and mortality [100].

Radiation and chemotherapy are also important therapeutic options [101]. However, radiation does not actually kill the cancer cells; instead, it causes DNA damage that cannot be repaired, causing the cells to stop dividing and eventually die. As a result, the cells are then expelled from the body [100]. In addition, chemotherapy is linked to serious side effects, damaging consequences for the skin, hair, blood, kidneys, and gastrointestinal system, and a higher risk of cancer post-therapy. Novel prospective therapeutic approaches are necessary, considering these shortcomings, to decrease the suffering and mortality brought on by malignancies [100,102,103].

The mainstay of conventional therapies for HNC is surgical resection, which is followed by radiotherapy and chemotherapy, with the specifics of the regimen depending on a variety of variables [104]. Chemotherapeutic drugs are used in OC either alone or in combination with radiation. The drugs used most are cisplatin, paclitaxel, cetuximab, docetaxel, fluorouracil (5-FU), or a combination of cisplatin and 5-FU [105].

Oral chemotherapy initially seems to have only positive effects. Patients spend more time at home rather than receiving a constant drug infusion over many hours in a healthcare center. Cancer patients have more freedom and are more self-sufficient, as they have better control over their treatment and don’t require medical supervision to administer the drug. Additionally, intravenous administration fees are eliminated [106,107]. The benefits of OC treatment are undeniable, and they essentially “boost” the wellbeing of patients who have been diagnosed with cancer and their families. Nonetheless, oral chemotherapy does pose additional risks, such as skin rash, hypertension, and thyroid dysfunction, that must be considered [108].

## 5. Natural Products in Oral Cancer and Their Limitations

The multimodal treatment for OC includes surgery, radiotherapy, chemotherapy, and immunotherapy. The mortality rate of HNSCC is still significant, despite the current clinical interventional techniques. In this regard, novel, more potent therapies with fewer side effects are required [109]. Chemoprophylactic therapies utilize natural and/or synthetic drugs to suppress, inhibit, or reverse OSCC. Preliminary evidence suggests a healthy diet can help prevent the development of OC, especially if the diet is comprised of many fruits and vegetables abundant in micronutrients such as β-carotene, vitamin C, vitamin D, and flavonoids [110,111,112]. Phytochemicals have been shown to have anticancer effects, primarily by controlling epigenetics/epigenomics [113], targeting cancer stem cells (CSCs) [114], impeding cancer metastasis [115], boosting immune function, antioxidation activity, and anti-inflammatory responses [116], hindering cell signal transduction [117], inhibiting cancer cell cycle progression, promoting cancer cell apoptosis [118], and finally succeeding in stopping angiogenesis and cancer cell growth [119,120,121]. However, phytochemicals often have poor water solubility, bioavailability, and targeting, which restrict their usage in therapeutic settings. Natural product delivery has major challenges due to the low solubility, poor absorption, inappropriate size, instability, poor permeation, and first-pass metabolism. To resolve these issues, various studies have examined the development of phytochemical delivery methods, and this continues to be a topic of interest today [26].

## 6. Formulation Strategies for Natural Products Targeting Oral Cancer: Mechanism and Bioavailability Enhancement

### 6.1. Nanostructural Systems

#### 6.1.1. NPs

The National Pharmacopoeia Council of China utilizes salvianolic acid B (SaIB), a bioactive constituent, as an active marker for products produced from *Salvia miltiorrhiza* Bge (Danshen). In fact, traditional Chinese medicine has employed the dried root of *Salvia miltiorrhiza* Bge (Danshen) to heal a wide range of ailments, including malignancy [122]. SalB phospholipid complex-loaded nanoparticles (SalB-PLC-NPs) have been tested for their efficacy on oral pre-cancerous and OSCC cell lines. In comparison to a free salvianolic acid formulation, varying concentrations of SalB-PLC-NPs effectively suppressed cell proliferation in both pre-malignant and malignant cells [123,124].

Polymeric NPs, particularly amphipathic block copolymer micelles, exhibit distinctive features that boost medication solubility and stability. This attribute of amphipathic block copolymer micellar system facilitated the development of an hydroxycampothecin (HCPT)-loaded nanoparticle system to address HCPT low solubility and stability issues, employing poly[ethylene glycol]-poly[gamma-benzyl-L- glutamate] (PEG-PBLG) as an amphipathic copolymer. In vitro cytotoxicity of this was assessed utilizing Tca8113 SCC cells, while in vivo experiments were conducted utilizing the golden hamster cheek pouch SCC model. Both investigations found that HCPT/PEG-PBLG micelles outperformed open-ring carboxylated HCPT with regard to anticancer activity [125].

Garlic intake has been shown to lessen the risk and progression of oral malignancy [126]. Allicin enhanced the catalytic activity of human salivary aldehyde dehydrogenase (hsALDH), which may lessen the likelihood of oral carcinogenesis [127]. At a concentration of 100 ng/mL, allicin reduced the cell survival of OSCC. It reduced TNF-α, IL-8, and endothelial protein expression [128]. Garlic extract-modified titanium dioxide NPs had cytotoxic action against OC cells, reducing cell viability with increasing dosage. At 10 mg/mL concentration, the percentage cell viability was determined to be 60.76%. Moreover, the generation of reactive oxygen species (ROS) resulted in a reduction in cell viability [129,130,131,132].

NPs loaded with naringenin (NAR) of (~90 nm size) were produced and designed for the OC treatment. NAR was successfully loaded in NPs, achieving a high encapsulation efficiency of 88 ± 2.7%, indicating 88% of the medication was encapsulated within them [133]. Finally, oral treatment of free NAR and NARNPs reversed the condition of lipid peroxidation and antioxidants in DMBA-painted animals’ buccal mucosal tissues. Free NAR and NARNPs have shown better antiproliferative activity by suppressing the expression of proliferating cell nuclear antigen (PCNA) and p53. In DMBA-induced hepatobiliary tract and pancreas (HBP) carcinogenesis, NARNPs were shown to possess greater anti-lipid peroxidative, antiproliferative, and antioxidant activity compared to free NAR [134].

The condition of bioactive components in the buccal mucosa of DMBA-painted hamsters was restored by oral delivery of free NAR and NARNPs. Ultimately, nanoparticulate NAR therapy was shown to be more efficient than free NAR in effectively avoiding the development of SCC and restoring many Raman bands to be within normal ranges in hamster buccal mucosa during DMBA-induced oral carcinogenesis. Nanoparticulate NAR exhibits great chemopreventive potential through its ability to limit or reduce aberrant cell growth in the buccal mucosa by disrupting the DMBA metabolic activation [135].

Numerous studies have shown that nanoparticulate drug delivery methods have higher antitumor activity while minimizing systemic toxicity [133,136]. The particle size is widely known to play an important influence in their interactions with cells and the in vivo destiny of a particulate system for drug delivery. Smaller particles appear to have more effective interfacial interactions with the cell membrane than larger particles, boosting the efficiency of particle-based oral drug delivery systems. Smaller particle size NPs (200 nm) have been shown to boost intratumor concentration of therapeutic agents by boosting permeability and retention (EPR) effects [137]. Such EPR effects are mostly caused by variations in blood vessels between tumor and normal tissues. Normal tissue vessels are bordered by compact endothelium that keeps NPs from entering the tissue, but tumor tissue vessels are abnormal, leaky, distended, and the endothelial cells are improperly positioned with wider fenestration. The EPR effect causes greater leakage of nanocarriers from the vasculature into tumor tissue as a result of this structure. Researchers hypothesized that the enhanced anticancer effect of NPs loaded with NAR relative to free NAR may be attributed to the following mechanism: when NAR is encapsulated in nanoparticulate systems, it can concentrate in tumor sites via the EPR effect and keep an efficient therapeutic concentration for a prolonged duration. This may result in increased antitumor effectiveness as well as alterations in biomolecular constitution as compared to free NAR [135].

NARNPs show significant antioxidant capability and free radical scavenging ability amid oral carcinogenesis when compared to NAR. Because of their huge surface/volume ratio, NARNPs provide numerous active sites for free radical scavenging, and nanoparticulate drug delivery devices can improve NARNP oral bioavailability. Furthermore, nanoparticulate carriers may contribute to better oral bioavailability by preventing NAR breakdown in the gastrointestinal tract, as well as improved intestinal absorption and protection against first pass metabolism. NPs loaded with NAR enhanced the antioxidant defense by scavenging overly produced ROS following DMBA-induced hamster buccal pouch malignancy. The improved activity of NARNPs might be attributed to differences in NAR bioavailability by NARNPs, which could explain the greater antiproliferative effect of nanoparticulate NAR. The increased bioavailability and stability of nanoparticulate NAR resulted in larger deposition of administered NAR inside tumor cells, resulting in more significant downregulation of PCNA and p53 than free NAR-treated animals [134]. The continuous exposure of tumor mass to released NAR from NPs might be one of the mechanisms driving NARNPs’ advantage over free NAR. In a prior work, researchers found that NARNPs had more sustained drug release than free NAR, potentially exposing tumor cells to anticancer agent for longer time period [133]. As a result, with NARNPs, drug nanoparticulates could target tumors via EPR effects and subsequently maintain an increased concentration steadily with time, resulting in better antitumor effectiveness when compared to free NAR [134].

Resveratrol NPs were created and investigated for potential anticancer and anti-inflammatory properties in cancer stem cells (CSCs). External stimuli were used to enhance the population of M1-like macrophages in co-cultured H357 OC cells and human leukemic monocyte cells (THP-1), and THP-1 cells alone. This stimulation boosted cytokine synthesis within cells. Subsequently, OC cells and patient-derived primary OC cells were incubated with cytokine-enriched H-357 + THP-1 cells and cytokine-enriched THP-1 cells, respectively, to generate a CSCs-enriched population. Resveratrol-NP diminished metastasis and angiogenesis by inhibiting the inflammatory cascade via reducing cytokine generation in an in vitro, ex vivo, and in vivo mouse xenograft model [138].

A PLGA-encapsulated nanoformulation of resveratrol NP was developed and described to augment the pharmacokinetics efficiency of resveratrol. Resveratrol inhibited cell growth in a concentration-dependent manner, with 50% cell growth inhibition (IC_50_) observed at 25 µM, whereas resveratrol-NP produced equivalent cell death at 5 μg/mL. This finding shows that resveratrol-NP was more efficient than free resveratrol in triggering cell death in H-357 cells. Cytokines produced from macrophages are accountable for controlling cancer stemness, metastasis, and angiogenesis; resveratrol-NP inhibited CSC proliferation, metastasis, and angiogenesis by suppressing cytokines in the CSC-abundant oral tumor cell microenvironment [138].

Several investigations into the toxicity of silica NPs (SiNP) have found that it can cause cytotoxicity by inducing oxidative stress in human bronchoalveolar carcinoma-derived cells [139,140,141,142]. Moreover, SiNP can limit replication, transcription, and cell proliferation by inducing abnormal aggresomal-like collections in the nucleoplasm after entering the cell nucleus [140]. Another analysis revealed that the cytotoxicity of silica (Si) to human cells is significantly dependent on their metabolic activity. Fibroblast cells having prolonged doubling times are more vulnerable to Si-induced damage compared to tumor cells with shorter doubling periods [142]. The higher phototoxicity of the drug-nanoparticle combination is related to improved uptake, as evidenced by cellular uptake as well as photostability, which clearly illustrate the benefit of utilizing SiNP as carriers. Since the 3-amino propyl functional group is cationic at physiological pH, it has been demonstrated before that organically modified SiNP containing vinyl and 3-amino propyl groups may attach to anionic photosensitizers via electrostatic force [143,144]. Although the electrostatic complex doubles the enhancement of absorption, the covalent complex rises by a factor of three. This implies that the real absorption of the Rose Bengal SiNP (RB-SiNP) covalent complex is more than indicated by the uptake assays [145].

Silibinin (SIL) is a plant-derived flavonoid found in silymarin, which has been isolated from the fruits and seeds of milk thistle (*Silybum marianum*) [146]. As a result, many SIL formulations have now been created to boost its solubility and hence bioavailability, such as beta-cyclodextrin inclusion complexes, phospholipid complexes, and polymeric NPs [147,148]. Eudragit^®^ E is a positively-charged copolymer that has been frequently utilized to increase the solubility of medications that are poorly soluble in water [149]. It has a basic site that contains tertiary amine groups that are charged in gastric fluid and dissolves easily in the gastrointestinal milieu [150]. This polymer combination allowed for careful manipulation of particle size and release profile (particularly the burst effect) compared to silibinin-loaded nanoparticles (SILNPs) control [151].

The increased cytotoxic effect of SILNPs might be attributed to their effective targeted binding and subsequent cell uptake. The cationic exterior of the complexes can aid NPs in binding closely to the anionic cellular membrane, enhancing endocytosis [152]. The sustained release of SIL from the nanoparticle formulation, as demonstrated by the in vitro drug release kinetics analysis, suggests drug dispersal out of the polymeric matrix of the NPs is required for efficient antiproliferative action [153]. The amorphous or disordered-crystalline condition of the drug within the preparation may be responsible for the SILNPs’ prolonged release action. SILNPs demonstrated antiproliferation action with a lower IC_50_ value of 15 g/mL compared to free SIL, which has an IC_50_ value of roughly 28 g/mL. This might be linked to the increased internalization of SILNPs into cells and their ability to bypass numerous drug resistance barriers [154,155]. The increased cytotoxic activity of SIL encapsulated in NPs compared to free SIL is also attributed to the drug’s constant exposure and continual release at the location of action for an extended duration. In the current work, SILNP-treated cells increased ROS formation 2.5 times faster than SIL-treated cells. These findings suggest that SIL administration of NPs allows for greater accumulation in cells, resulting in increased intracellular ROS production. The higher mitochondrial membrane potential (MMP) modification in SILNP-treated cells compared to free SIL treatment suggests that NPs release SIL intracellularly in a direct and regulated manner [156].

Considerable apoptosis-related morphological modifications, including apoptotic body development and chromatin condensation, were observed in cancer cells treated with SILNPs. This might be due to increased internalization of drug-loaded NPs into cells, which causes apoptosis. In the current work, SILNP treatment resulted in substantial DNA damage, as evidenced by the creation of a comet. The percentages of tail length, tail DNA, tail moment, and olive tail moment in the control, free SIL, and SILNPs were used to determine the degree of DNA damage. When compared to the free SIL, the treatment with SILNPs results in a much higher percentage of tail length, tail DNA, tail moment, and olive tail moment in KB cells. The enhanced damage of DNA in SILNPs might be attributed to higher ROS production [156].

#### 6.1.2. Microemulsions

Curcumin microemulsions were assessed for their cytotoxic effects on tongue tumor cells, both with and without low-frequency ultrasound in reference to a free curcumin preparation. In comparison to a pure curcumin formulation, curcumin microemulsions demonstrated increased cytotoxicity, and the utilization of ultrasound significantly reduced cell viability [124,157].

#### 6.1.3. SLN

Andrews et al. investigated the absorption and retention of SLNs containing the anticancer drug in human OSCC cell lines SCC4, SCC9, SCC15, and SCC25. The findings suggest that the SLN-based therapeutic approach increases SLN penetration and intracellular levels in proliferating SCC cells of the basal layer [158].

#### 6.1.4. Niosomes

Curcumin is a potent antioxidant and has a wide array of biological and pharmacological properties. The sole drawback of using this chemical is its poor solubility, which leads to limited stability and bioavailability. In order for this substance to deliver the intended therapeutic benefits, it must be administered fairly frequently and in substantial doses [159]. Niosome nanocapsules are transporters generated by the aggregation of non-ionic surfactants in an aqueous medium and subsequently formed into a vesicle-like double-layered structure. Furthermore, niosome nanocapsules are divided into both a hydrophilic and a hydrophobic part. The usage of curcumin-loaded niosomes enhances the pharmacokinetics of the delivered medications, increasing the therapeutic benefits and minimizing negative effects. Stability of the drug might be improved as a result of the drug’s accumulation within the noisome [160,161].

Curcumin encapsulation efficacy in niosomes was evaluated by comparing it to the non-encapsulated drug. The findings revealed that 98% of the curcumin was incorporated into niosomes. According to overall assessment and microscopy analysis, both mouthwash and injectable types of curcumin-loaded niosomes increased the rats’ ability to prevent progression of OC when compared to the negative control group. Curcumin-loaded niosomes are substantially more efficacious than free curcumin at preventing cancer cell proliferation and necrosis. The curcumin niosome system was shown to be successful and effective in preventing rat OC, demonstrating the niosome approach successfully displays favorable results. A dosage of 16 g after 24 h was chosen as an appropriate dose at the cellular level. Curcumin-loaded niosomes were found to be efficient in avoiding the occurrence of severe dysplasia and inhibiting cancer cell proliferation in these in vitro and in vivo investigations [161].

### 6.2. Site-Specific and Target-Oriented Delivery Systems

#### 6.2.1. Gels

The black raspberries are abundant in vitamins, minerals, fiber, anthocyanins, phenolic compounds, and other bioactive ingredients with anticancer properties [162], and previous findings suggest they can prevent a number of malignancies, including OC [163]. Mucoadhesive gels comprised of Noveon AA1 and Carbopol 971 polymers have been produced and studied for the site-specific intraoral administration of black raspberry anthocyanins (BRAs), which display chemopreventive effects [164,165,166,167,168]. The pH of the gels and storage temperature had a substantial influence on the chemical stability and mucosal penetration of BRAs [164]. The mucoadhesive gels exhibited optimal stability of BRAs. Following incubation of gels with human oral mucosa explants, anthocyanins quickly diffuse into the human oral mucosa [164].

#### 6.2.2. Microspheres

A key constituent in the *Cannabis sativa* L. (marijuana) family of plants is cannabidiol, and in vivo research shows that it is an efficient cytotoxic agent against HNSCC and suppresses the progression of head and neck tumors [165]. The local treatment of D9-tetrahydrocannabinol and cannabidiol-encapsulated polycaprolactone (PCL) microspheres daily for 5 days decreased tumorigenesis to the same extent as daily localized administration of the comparable dose of the cannabinoids in solution [169]. Specifically, the PCL microsphere-based local administration to the murine xenograft model increased apoptosis while decreasing cell proliferation and angiogenesis in malignancies [169].

#### 6.2.3. Nanoliposomes

A study was conducted to understand viability of nanoliposomes embedded in thermoreversible Pluronic F127 hydrogel as an injectable formulation for paclitaxel sustained administration [170]. The authors produced drug-loaded nanoliposomes using a solvent dispersion approach. When compared to TaxolVR, in vitro release evaluations revealed more regulated delivery of paclitaxel from nanoliposomes and nanoliposomes/hydrogel (the commercial formulation). As opposed to the nanoliposomal formulation, the liposomal-hydrogel formulation demonstrated enhanced cytotoxic effects and drug concentration, suggesting that incorporating nanoliposomes in the hydrogel matrix can provide a clinical benefit [170].

#### 6.2.4. In Situ Gels

A temperature-sensitive mucoadhesive Sol-Gel system to administer the anticarcinogenic medication paclitaxel utilizing the temperature-sensitive polymer pluronic F127 and the mucoadhesive polymer polyethylene oxide was developed [171]. Paclitaxel’s water solubility was enhanced using dimethyl-cyclodextrin. This Sol-Gel formulation amplified paclitaxel in vitro release and cytotoxicity. This novel technique can be employed as a buccal mucoadhesive platform for localized administration of anticancer medicines to treat OSCC [172].

#### 6.2.5. Hydrogels

Thermosensitive chitosan hydrogel with paclitaxel-loaded nanoliposomes was developed as a transporter system as a two-fold means of targeted and sustained administration of paclitaxel to the location of the malignancy [173]. This strategy revealed longer release of the drug for more than 72 h when compared to the marketed formulation, as well as increased half-life to 15.7 ± 1.5 h when compared to the marketed formulation’s half-life of 3.6 ± 0.4 h. Tumor volume was decreased by up to 89.1%. The results suggest this hydrogel-based carrier system may be employed to treat OSCC locally [172,174].

#### 6.2.6. Nanoemulsions (NE)

NE comprised of mixed-polyethoxylated emulsifiers and a tocopheryl moiety-enriched oil phase as lipid-based nanocarriers were formulated. Prototype Nes were then manufactured into buccal tablet formulations incorporating the proapoptotic lipophilic substance. By functioning as a mucoadhesive interfacial NE layer, the chitosan polyelectrolyte solution coating converted NE droplets into their cationic forms. The positively charged chitosan-layered NE (+25 mV) displayed a controlled-release profile and efficient mucoadhesion for liquid oral spray prototypes, with an estimated size of 110 nm compared to negatively charged chitosan-free/primary aqueous NE (−28 mV). In compression testing, chitosan-containing NE tablets were shown to be similar to original NE and placebo tablets, but displayed more favorable outcomes in all ex vivo adhesion and in vitro release studies. Following biocompatibility testing of prototype chitosan-layered NEs, significant anticancer efficacy of certain cationic genistein-loaded NE formulations was reported against two oropharyngeal carcinomas. The findings significantly support the use of nanomucoadhesive systems as sustained treatment for patients with upcoming OC surgical removal or post-resection of detected malignant lesions [175].

The chitosan-coated nanoemulsified technology (in liquid or tablet dosage form) could not only improve anticancer drug partitioning to the oral mucosal barrier, but also facilitate the targeted release of genistein to the localized area of action, i.e., precisely into malignant lesions in the mouth, pharynx, and tongue [175]. Because of the capacity to objectively examine and modify the interfacial features of our chito-layered NE systems, improved drug-loaded droplet formulations were created with increased stability and mucoadhesive functional capabilities. The further production of proof-of-concept buccal liquids and tablets culminated in prolonged distribution of the antiproliferative therapeutic agent, genistein, with specific anticancer properties. Evidence from the innovative mucoadhesive buccal sprays and lozenges indicate that such platforms have the ability to be employed as adjuvant treatment for patients with OC [175].

#### 6.2.7. Mucoadhesive NPs

Curcumin-loaded mucoadhesive NPs were created as a novel method of administering curcumin for the local treatment of OC. The nanoprecipitation approach was used to create polycaprolactone (PCL) NPs coated with chitosan with varying molecular weights [176]. Chitosan’s cationic polyelectrolyte nature creates a strong electrostatic contact with the negatively charged mucosal surface, which may enhance the medication device’s duration at the absorption site. Furthermore, chitosan has been demonstrated to enhance structural remodeling of epithelial cell tight junction-associated proteins, which may boost mucosal medication transport [177,178].

The ability of nanoparticle suspensions to interact with the glycoprotein mucin via electrostatic interactions demonstrates their mucoadhesive capabilities. Curcumin concentrations preserved in the mucosa suggest that the medicine may have a local impact. In vitro experiments revealed that free curcumin and curcumin loaded into chitosan-coated NPs markedly reduced SCC-9 human OC cell survival in a concentration and time-dependent manner. However, following treatment for 24 h with unloaded NPs coated with chitosan, no substantial cancer cell death was observed. Furthermore, as compared to the free medication, curcumin-loaded NPs exhibited lower normal cell cytotoxicity. As a result, chitosan-coated PCL NPs might be a potential technique for administering curcumin directly into the mouth cavity to treat oral tumors [176].

Curcumin has been shown to decrease numerous OSCC cell development by upregulating insulin-like growth factor binding protein 5 (IGFBP-5) and CCAAT/enhancer binding proteins (C/EBPs), as well as decreasing human OSCC cell motility by suppressing NF-κB activation [179,180]. Recent research using the OSCC cell line CAL-27 demonstrates that curcumin’s anticancer action is mediated by a new mechanism involving the inhibition of the Notch-1 and NF-κB signaling pathways [181]. Finally, NPs exhibited a curcumin concentration of around 500 µg/mL and encapsulation efficiency values over 99%, confirming their potential for curcumin incorporation [176].

Free curcumin had the greatest cytotoxicity impact. The total reduction in viable cells following incubation for 72 h with free curcumin was approximately 90%, whereas that of curcumin-loaded NPs reduced around 45%. This impact might be explained by the medicine being encapsulated within the NPs, which makes the drug less accessible to interact with the cells. Due to this, the dosage may be modified to provide a pharmacological effect, and the period of effect might be lengthened by managing release of the drug. Moreover, encapsulating curcumin in polymeric NPs appears to be advantageous since it enables the delivery of a hydrophobic medication as an aqueous dispersion, and this formulation might increase bioavailability and reduce its hydrolytic and photochemical degradation [176].

A detailed insight on the effect of bioavailability of various natural product-based nanoformulations is presented in Table 1 and Figure 3.

## 7. Conclusions and Future Perspectives

OC has become one of the most prevalent malignancies with high mortality rates globally. Chronic consumption of tobacco products and alcohol has been associated with a high incidence of fatalities. Conventional treatments available for the treatment of OCs include chemotherapy, radiotherapy, and surgery. These conventional methods lack optimal antitumor effects and exhibit non-specific cell toxicity. Alternative therapeutic approaches might resolve the concerns and shortcomings associated with conventional treatment strategies. Additionally, the cytotoxicity of presently available anticancer medicines is a substantial obstacle in the treatment of OC. Nonetheless, using natural substances to prevent cancer may limit the related toxicity. Overall, this review summarizes various natural products that target several signaling pathways implicated in the growth of oral tumor cells, signifying their possibility as potent antitumor drugs (Figure 4). Natural products are typically constrained by their limited bioavailability and poor targeting; however, novel drug delivery carriers can be designed to mitigate these issues.

Nanotechnology-based drug delivery systems have been extensively explored and are emerging as potential alternative carriers for the treatment of a variety of malignancies. Various novel drug delivery approaches utilizing nano-structured and site-specific targeting systems of natural products for the treatment of OC are summarized in this review (Table 2). The various studies discussed above conclude that these novel systems enhance the bioavailability of natural compounds through various mechanisms, such as EPR, particle size reduction, mucoadhesion, and drug encapsulation. In addition to bioavailability augmentation, such novel systems also offer other benefits including low dose, reduction of dosage frequency, decreased drug resistance, and enhanced patient compliance with an overall improvement in the treatment of OC. However, in vivo bioavailability data was lacking in most of the studies conducted on novel formulations containing natural products, thus posing a limitation on the determination of treatment efficacy. The challenges faced in the review and study of literature were the lack of pharmacokinetics, in vitro, and in vivo correlation studies on natural products and traditional medicines, as well as the limited number of clinical trials to show their bioavailability. More research is required to adapt nanotechnology principles into potential practical implementation, which will aid in determining optimum therapeutic dosages and methods for efficient drug release at target sites for the treatment of various tumors. Consequently, it is recommended that further exploration, in particular clinical studies, be performed on natural products and their formulations. This would provide a better understanding of the dosage regimen to advise the treatment of cancer in the future.

## Figures and Tables

**Figure 1 cancers-15-00268-f001:**
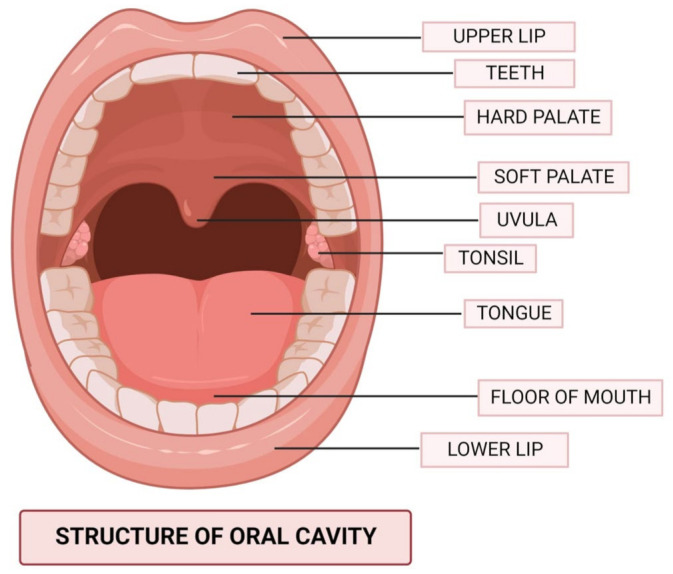
The anatomical structure of the oral cavity.

**Figure 2 cancers-15-00268-f002:**
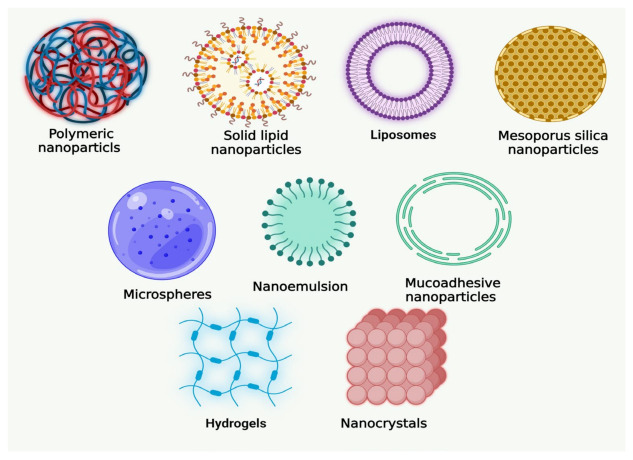
Various novel formulations of natural products developed for oral cancer.

**Figure 3 cancers-15-00268-f003:**
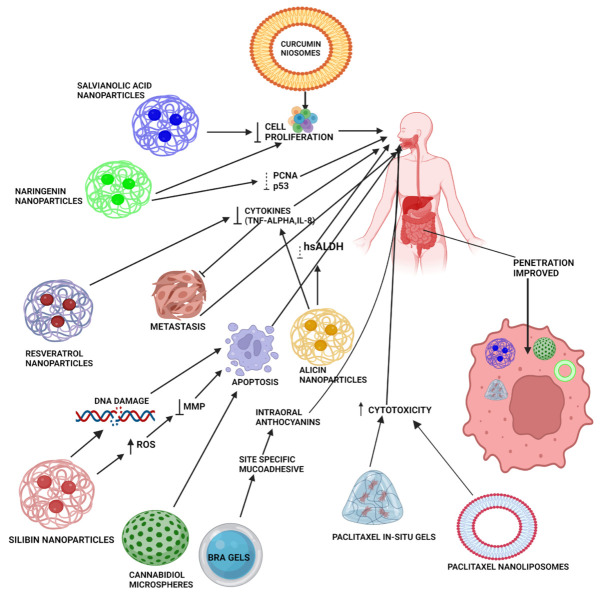
Mechanistic illustration of various novel formulations (NPs, in situ gels, microspheres, hydrogels, niosomes, and nanoliposomes) of natural products (curcumin, paclitaxel, alicin, resveratrol, BRAs, SIL, NAR, and salvianolic acid) for the treatment of oral cancer. These advanced formulations enhance penetration of natural products by acting through different pathways for the suppression of oral cancer progression, including reducing mitochondrial membrane potential, metastasis, various cytokines, such as TNF-α, IL-8, and activity of various enzymes, such as hsALDH. All these effects ultimately cause DNA damage and lead to apoptosis, thereby reducing the rate of oral cancer cell proliferation. Various gels provide site-specific delivery of natural products, such as paclitaxel, by incorporation of mucoadhesive polymers, leading to enhanced bioavailability at the site of action.

**Figure 4 cancers-15-00268-f004:**
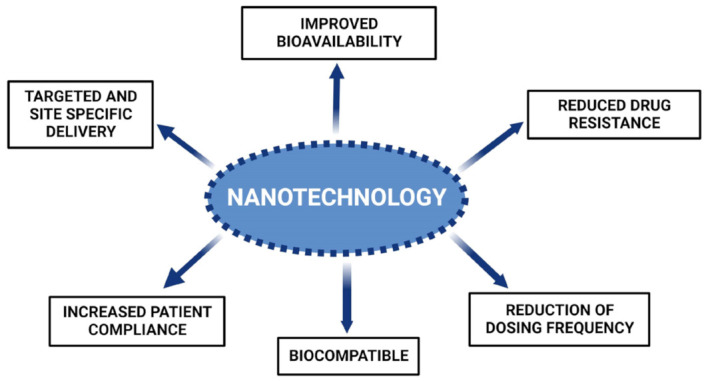
Numerous advantages offered by nanosystems of natural compounds for oral cancer.

**Table 1 cancers-15-00268-t001:** Novel drug delivery system-based formulations of natural products for oral cancer.

Natural Products/Extract	Formulation Type	Dose/Conc.	Polymer Used	Bioavailability /In Vitro Release	Major Outcome	References
BRAs	Gel	Applied 10% (*w*/*w*) freeze dried black raspberry gel (0.5 g gel) four times/day for 6 weeks.	Noveon AA1 and Carbopol 971 polymers	NR	Reduced lesion size and microscopic histological grade in 35% of patients	[164]
Camptothecin	Polymeric nanoparticles	33–40 microg/mL	Cyclodextrin derivative chitosan	47–51% drug content	Increased oral bioavailability	[182]
	NE	NR	Poloxamer 188	NR	Overcame the solubility and stability	[183]
Curcumin	Mucoadhesive NPs	500 µg/mL	PCL + chitosan	NR	Improved bioavailability, decreased hydrolytic and photochemical degradation of curcumin	[176]
	Nanoniosomes	16 μg	NR	NR	Overcame poor oral bioavailability, enhanced drug stability	[161]
D9 –Tetrahydrocannabinol, Cannabidiol	Microspheres	(9 wt%- drug loading)	PCL	NR	Reduced tumor growth by enhanced apoptosis and decreased cell proliferation and angiogenesis	[169]
Genistein	NE	2 mg/mL	Chia seed oil, DL-a-tocopherol	NR	Enhanced anticancer drug partitioning to oral mucosal membrane, targeted delivery of Genistein at the site of action	[175]
Naringenin	NPs	50 mg	EE:PVA	High encapsulation efficiency of 88 ± 2.7%	More potent anti-lipid peroxidative, antiproliferative and antioxidant potentials	[134]
Paclitaxel	Mucoadhesive Sol-Gel system	0.2 mg	Thermosensitive polymer pluronic F127 and mucoadhesive polymer polyethylene oxide	90% of the drug over 3-day period; moreover, the release was sustained	Improved paclitaxel solubility, reduced toxicity	[184]
	Hydrogel	6 mg/mL	Chitosan	Drug release was found to be 32.3 ± 1.3% in 24 h and 61.7 ± 2.6% in 72 h	Tumor volume was reduced up to 89.1 ± 3.5%	[173]
	NPs	150–230 mg/m^2^ frequency: 2–4 infusions every 3 weeks.	Albumin	NR	Clinical and radiologic objective response in the majority of patients (78%). Intraarterial infusion of paclitaxel in albumin nanoparticles proved reproducible and effective	[185]
	Nanoliposomes hydrogel	300 μg/mL	Lipids [lipids containing soya phosphatidylcholile, nitro benzoxadiazol-labeled phosphatidylethanolamine]	NR	Exhibited greater cytotoxicity and provide a higher drug concentration	[170]
Resveratrol	NPs	5 µg/mL	PLGA-PEG-COOH	NR	Res-NP reduced the CSCs growth, metastasis, and angiogenesis by inhibiting the cytokines in CSCs enriched oral cancer cells niche	[138]
Rose Bengal	Si NPs	(For Preparation—o 400 microM RB)	Aerosol AT	NR	Enhanced phototoxicity by enhanced uptake	[145]
SIL	NPs	15 μg/mL	EE 100, PVA	24.1% of the entrapped SIL release in 6 h) (~79.2% of the drug released in 24 h	Enhanced cytotoxicity of SILNPs extensive DNA damage, increased MMP alteration	[156]
Garlic	Garlic extract-modified titanium dioxide NPs	10 mg/mL	NR	60.76%	Exhibited cytotoxic activity against oral cancer cell line by decreasing the cell viability; the production of ROS led to decrease in cell viability	[129]

Abbreviations: BRAs, black raspberry anthocyanins; CSCs, cancer stem cells; EE, eudragit E; MMP, mitochondrial membrane potential; NE, nanoemulsion; NPs, nanoparticles; PCL, polycaprolactone; PLGA-PEG-COOH, poly(lactide-co-glycolide)-block-poly(ethylene glycol)-carboxylic acid; PVA, polyvinylalcohol; Res, resveratrol; ROS, reactive oxygen species; SIL, silibinin.

**Table 2 cancers-15-00268-t002:** Comparative analysis of bioavailability of pure natural products and their advanced formulations.

Natural Products/Extract	Strategy for Bioavailability Augmentations	Conc./ Dose	Bioavailability Enhancement (Effect of ADMET)		Effect on Oral Cancer	Mechanisms	Reference
			Bioavailability of Normal Drug	Bioavailability of Novel Formulation			
Capsaicin	SNEDDS	305.41 g/mol	16.61 ± 3.64%	3.6-fold increase in bioavailability	Antiproliferative effects	MMP disruption, caspase-3, caspase-7 and caspase-9 activation through an intrinsic apoptotic pathway and subsequently, apoptotic DNA fragmentation	[186]
EGCG	NE	200–800 mg	NR	The bioavailability was more than 2.78-fold	Inhibition of both cell proliferation and migration	Targets multiple signaling pathways, including the downregulation of EGFR and associated downstream signaling molecules	[187,188,189,190]
Piperlongumine	NPs	7.4–11.3 μM	NR	NR	Antiproliferative effects, cell cycle arrest and senescence	↓PI3K/Akt/mTOR pathway, ↓ROS, ↑apoptosis, ↑G1 phase cell cycle arrest, ↑p21, ↑cleaved caspases-3, ↑PARP	[191,192]
Bromelain	Lipid-polymer hybrid nanoparticles	12.5–100 µg mL	NR	Maximum release of Bromelain from nanocarriers was obtained 30–35% after 5 days	Inhibited cell growth and proliferation	G1 cell cycle arrest, induced apoptosis	[26,193,194,195]
Curcumin	Curcumin nanocrystals/NPs	10 or 12 g/mL	1%	Over 5 times from simple curcumin powder	Inhibit growth, invasion and metastasis	Inhibits the invasive ability and EMT by reducing the MMP-2 and MMP-9 expression, downregulation of EGFR expression	[196]
Berberine	Vitamin E d-α-tocopheryl polyethylene glycol 1000 succinate-mixed polymeric phospholipid micelles of berberine	100 or 300 mg/kg	<5% in plasma	15%	Antimitotic and proapoptotic actions, along with distinct antiangiogenic and antimetastatic activities	Suppresses the mRNA expression of NFKB1 and PTGS2 and AURKA, BIRC5, and EGFR	[26,197]
Honokiol	NPs	0–150 mg/kg	NR	53% of honokiol was released from the nanoparticles within 24 h	Antiproliferative effect	Blocks EMT through the modulation of Snail/Slug protein translation	[198]
Evodiamine	Nanocomposite system comprising folic acid- modified graphene quantum dots	1 mg	NR	Over 90% of drug was released in 72 h	Inhibited cell proliferation	Downregulates Mcl-1 expression, induces apoptosis mediated by a caspase-dependent pathway	[199]
Gedunin	Chitosan- encapsulated gedunin	1.5–50 µg/mL	NR	NR	3 to 8-fold decrease	Modulates AR, PI3K/Akt, and NF-κB pathways to block angiogenesis	[200]
Sinularin	Not found	NR	NR	NR	Selectively kill the oral cancer cells	Antiproliferative and apoptotic effects on oral cancer cells coupled with ROS generation and G2/M arrest	[201]

Abbreviations: AR, aldose reductase; AURKA, aurora kinase A; EGCG, epigallocatechin gallate; EGFR, epidermal growth factor receptor; EMT, epithelial-mesenchymal transition;; MMP, mitochondrial membrane potential; NF-κB, nuclear factor-κB; PARP, poly adenosine diphosphate-ribose polymerase; PI3K, phosphatidyl inositol-3-kinase; PTGS2, prostaglandin coding gene; ROS, reactive oxygen species; SNEDDS, self-nano emulsifying drug delivery system.

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
