# Peer review of "Novel Strategies for the Bioavailability Augmentation and Efficacy Improvement of Natural Products in Oral Cancer"

_cancers, 2022, doi:10.3390/cancers15010268_

Round 1

Reviewer 1 Report

The present review manuscript titled “Novel Strategies for the Bioavailability Augmentation and Efficacy Improvement of Natural Products in Oral Cancer” by et al is very well written. This manuscript is novel and the authors cover almost all the aspects. However, this manuscript needs revision. My comments are as follows.

Comment 1. First of all, the title should be revised. The title is not catchy.

Comment 2. In the present manuscript, different nanoparticles are discussed then why the authors provided a diagram of only 6 nanoparticles. The authors should provide the image of each nanoparticle that is discussed in the manuscript.

Comment 3. The authors discussed the different nanoparticles in two sections. The authors should discuss the nanoparticles in one section only and then the authors should discuss the phytochemical-loaded nanoparticles in the management of oral cancer in another section.

Comment 4. The abbreviation used in the manuscript is incorrect. Kindly revise from top to bottom.

Comment 5. References are full of mistakes. Kindly revise manually.

Author Response

The authors of this manuscript express their sincere thanks to the reviewer for the critical assessment of this work. The authors have acted upon the recommendations of the reviewer which have resulted in a significant enhancement in the quality of this manuscript. All modifications incorporated in the manuscript are highlighted in a red color font. A “point-by-point” response to each comment is outlined below.

General comments:

The present review manuscript titled “Novel Strategies for the Bioavailability Augmentation and Efficacy Improvement of Natural Products in Oral Cancer” by et al is very well written. This manuscript is novel and the authors cover almost all the aspects. However, this manuscript needs revision. My comments are as follows.

Response:

We would like to thank the erudite reviewer for his/her appreciation of the manuscript. We have tried our best to address the comments raised by the reviewer and revised our manuscript.

Specific comments:

Comment 1:

First of all, the title should be revised. The title is not catchy.

Response:

With due respect to the esteemed reviewer, we think the present title precisely and clearly captures the essence of the manuscript.

Comment 2:

In the present manuscript, different nanoparticles are discussed then why the authors provided a diagram of only 6 nanoparticles. The authors should provide the image of each nanoparticle that is discussed in the manuscript.

Response:

We thank the reviewer for this excellent comment and accordingly, we have modified the diagram (Figure 2) which now incorporates all the nanoparticles discussed in the manuscript.

Comment 3:

The authors discussed the different nanoparticles in two sections. The authors should discuss the nanoparticles in one section only and then the authors should discuss the phytochemical-loaded nanoparticles in the management of oral cancer in another section.

Response:

We appreciate this constructive recommendation. We have merged previous sections 6 and 7 to present a unified section (now section 6). The following changes are highlighted in red color font:

Section 6.1.1: Page 10, line 437 to page11, line 475

Section 6.1.1: Page 11, lines 486-510

Section 6.1.1: Page 11, line 522 to page 12, line 550

Section 6.1.4: Page 13, lines 577- 588

Section 6.2.6: Page 14, lines 662-673

Section 6.2.7: Page 15, lines 694-713

Comment 4:

The abbreviation used in the manuscript is incorrect. Kindly revise from top to bottom.

Response:

We are sorry for the errors with abbreviations. All the abbreviations are revised carefully throughout the manuscript and updated ones are highlighted in a red color font. A list of abbreviations used in the manuscript has been provided at the end of the manuscript (pages 20 and 21).

Comment 5:

References are full of mistakes. Kindly revise manually.

Response:

We sincerely apologize for our oversight. All the references are revised carefully following the journal style.

Additionally,

  1. The entire manuscript has been thoroughly checked and edited to minimize typographical errors as well as to ensure uniform style, organization, and quality.
  2. The reference list has been modified as we have deleted several new references. Special attention is given to conform to the order of references and bibliographic style of the journal.

Finally,

On behalf of my co-authors, I once again express my sincere thanks to the erudite Assistant Editor and reviewers for the valuable suggestions and constructive input to improve the quality of our manuscript.

Reviewer 2 Report

The authors of the article “Novel Strategies for the Bioavailability Augmentation and Efficacy Improvement of Natural Products in Oral Cancer” have given a comprehensive overview on the use various nanoparticle-based products for enhancing natural product delivery. There are some major concerns which needs to be addressed before this review can be deemed suitable for publication.

1.      The review has a lot of extra information which is somewhat out of the context e.g- the first paragraph of the Introduction describing the prevalence of cancer in general is unnecessary and can be eliminated from the review.

2.      In Figure 2, the nanoparticles described are not limited to oral cancer-they are used in general in different types of cancer, as suggested in Reference 44,45. The authors should refer and make figures specifically for different types of Oral cancer.

3.      Epidemiology in Section 2 has been already touched upon on Section 1 (Introduction). So, this section can be reduced or even eliminated.

4.      In many instances, the review digresses from the main topic of discussion and represents prior publications of natural products used for cancer in general rather than OC specifically. E.g- Resveratrol in Ref 144-146 has been discussed in detail as an anti-inflammatory agent for cancer in general. Similarly, section 6.1.3 describes SLNs against A549 cells, which are essentially lung cancer cells. This should be restructured and designed specifically for OC cell lines.

5.      Section 6 and 7 can be merged, since there is a lot of re-iteration of the same information.

Author Response

The authors of this manuscript express their sincere thanks to the reviewer for the critical assessment of this work. The authors have acted upon the recommendations of the reviewer which have resulted in a significant enhancement in the quality of this manuscript. All modifications incorporated in the manuscript are highlighted in a red color font. A “point-by-point” response to each comment is outlined below.

General comments:

The authors of the article “Novel Strategies for the Bioavailability Augmentation and Efficacy Improvement of Natural Products in Oral Cancer” have given a comprehensive overview on the use various nanoparticle-based products for enhancing natural product delivery. There are some major concerns which needs to be addressed before this review can be deemed suitable for publication.

Response:

We would like to thank erudite reviewer for his/her appreciation of the manuscript. We have tried our best to address the specific comments and revised our manuscript.

Specific comments:

Comment 1:

The review has a lot of extra information which is somewhat out of the context e.g- the first paragraph of the Introduction describing the prevalence of cancer in general is unnecessary and can be eliminated from the review.

Response:

We appreciate this useful suggestion. The irrelevant information (e.g., the previous first paragraph) has been removed from the introduction.

Comment 2:

In Figure 2, the nanoparticles described are not limited to oral cancer-they are used in general in different types of cancer, as suggested in Reference 44,45. The authors should refer and make figures specifically for different types of Oral cancer.

Response:

We sincerely appreciate this comment. Figure 2 has been modified and all the nanoparticles for oral cancer have been included in the figure.

Comment 3:      

Epidemiology in Section 2 has been already touched upon on Section 1 (Introduction). So, this section can be reduced or even eliminated.

Response:

We agree with this suggestion. The section on epidemiology has been eliminated.

Comment 4:

In many instances, the review digresses from the main topic of discussion and represents prior publications of natural products used for cancer in general rather than OC specifically. E.g- Resveratrol in Ref 144-146 has been discussed in detail as an anti-inflammatory agent for cancer in general. Similarly, section 6.1.3 describes SLNs against A549 cells, which are essentially lung cancer cells. This should be restructured and designed specifically for OC cell lines.

Response:

We believe the reviewer has made an excellent observation. We are in absolute agreement with the reviewer. We have deleted various statements on other cancer types and focused only on oral cancer.

Comment 5:

Section 6 and 7 can be merged, since there is a lot of re-iteration of the same information.

Response:

We thank the reviewer for this important suggestion and accordingly sections 6 and 7 have been merged into a single section (section 6, title Formulation Strategies for Natural Products Targeting Oral Cancer: Mechanism and Bioavailability Enhancement, page 9, line 389 to page 15, line 713).

The following changes are highlighted in red color font:

Section 6.1.1: Page 10, line 437 to page11, line 475

Section 6.1.1: Page 11, lines 486-510

Section 6.1.1: Page 11, line 522 to page 12, line 550

Section 6.1.4: Page 13, lines 577- 588

Section 6.2.6: Page 14, lines 662-673

Section 6.2.7: Page 15, lines 694-713

Additionally,

  1. The entire manuscript has been thoroughly checked and edited to minimize typographical errors as well as to ensure uniform style, organization, and quality.
  2. The reference list has been modified as we have deleted several new references. Special attention is given to conform to the order of references and bibliographic style of the journal.

Finally,

On behalf of my co-authors, I once again express my sincere thanks to the erudite reviewer for the valuable suggestions and constructive input to improve the quality of our manuscript.

Round 2

Reviewer 1 Report

The authors revised the manuscript very carefully as per the comments. I don't have further comments.

Reviewer 2 Report

The manuscript has been sufficiently revised. It is now fir for publication.